# The Intelligible and Effective Graph Neural Additive Networks

**Maya Bechler-Speicher**
Blavatnik School of Computer Science
Tel-Aviv University

**Amir Globerson**
Blavatnik School of Computer Science
Tel-Aviv University*

**Ran Gilad-Bachrach**
Department of Bio-Medical Engineering
Edmond J. Safra Center for Bioinformatics
Tel-Aviv University

## Abstract

Graph Neural Networks (GNNs) have emerged as the predominant approach for learning over graph-structured data. However, most GNNs operate as black-box models and require post-hoc explanations, which may not suffice in high-stakes scenarios where transparency is crucial. In this paper, we present a GNN that is interpretable by design. Our model, Graph Neural Additive Network (GNAN), is a novel extension of the interpretable class of Generalized Additive Models, and can be visualized and fully understood by humans. GNAN is designed to be fully interpretable, offering both global and local explanations at the feature and graph levels through direct visualization of the model. These visualizations describe exactly how the model uses the relationships between the target variable, the features, and the graph. We demonstrate the intelligibility of GNANs in a series of examples on different tasks and datasets. In addition, we show that the accuracy of GNAN is on par with black-box GNNs, making it suitable for critical applications where transparency is essential, alongside high accuracy.

## 1 Introduction

In many domains, ranging from biology to fraud detection, Artificial Intelligence (AI) is applied to data with graph structure. Neural Networks, and specifically Graph Neural Networks (GNNs), have emerged as the predominant approach in these applications (see, for example, Zhou et al. [1]). While GNNs demonstrate high accuracy, in terms of the correctness of their predictions, they often function as black-box models; thus, their decision-making processes are opaque. Transparency is vital for assessing potential biases or safety risks, and is particularly critical in high-stakes areas such as criminal justice, healthcare, and finance, where decisions significantly impact individuals' lives. In such contexts, interpretable models, despite sometimes being less accurate, may be preferred over complex black-box models [2]. Furthermore, the transparency of automated decision making processes is increasingly becoming a legal mandate. While there is ongoing debate over whether the European Union's General Data Protection Regulation (GDPR) implies a *"right to explanation"* [3, 4], the proposed European AI Act explicitly addresses this issue, stating that *"To address concerns related to opacity and complexity of certain AI systems and help deployers to fulfill their obligations under this regulation, transparency should be required for high-risk AI systems before they are placed on the market or put into service"* [5].

---

*Now also at Google Research

38th Conference on Neural Information Processing Systems (NeurIPS 2024).

In this context, interpretability refers to the ease with which a human can understand the reasoning behind model decisions or the general logic of a model's operation. It is important to distinguish between interpretability and explainability [2]. Interpretability relates to models that are inherently comprehensible by design, while explainability pertains to post-hoc methods that elucidate aspects of black-box models [6]. These explanations often come without correctness guarantees [7, 8] and may not provide a complete description of the model and its predictions, potentially failing to expose hidden pitfalls [9, 10, 11].

Methods for model explainability or interpretability can be categorized into local and global types. Local methods, such as SHAP [12] and LIME [13], elucidate individual predictions made by a model, whereas global methods, such as feature-importance [14] and partial dependence plots [15], provide holistic insights about the model, i.e., explain the overarching logic of the model decision-making [16]. However, it has been noted that local explainability methods may not consistently align with their global counterparts [17]. Moreover, local explanations may be inadequate for verifying fairness and other risks [8].

In this work, we introduce the Graph Neural Additive Networks (GNAN), an interpretable-by-design GNN that offers both transparency and accuracy. GNAN is a glass-box model [18] that allows for both local and global interpretability. GNAN extends the family of Generalized Additive Models (GAMs) [19], to accommodate graph data. GAMs are known for their ability to fit complex, nonlinear functions while remaining interpretable and have proven effective across various domains [20, 21, 22, 23, 24]. They operate by learning shape functions for each feature and then linearly combining these functions, making it easy to interpret them, as the influence of each feature on the prediction is independent of other features and can be visualized through their corresponding shape functions. Similarly, GNAN's interpretability is achieved through an architecture that restricts the use of cross-products of features and graphs' topology, thereby reducing its complexity compared to other GNNs. Nonetheless, we demonstrate that GNAN, despite its limited capacity, matches the performance of more expressive GNNs on several real-world datasets. Additionally, GNAN does not rely on iterative local message-passing, avoiding the computational bottlenecks commonly associated with such GNNs [25].

In Section 4, we showcase through a series of examples how users can interpret GNAN and gain precise insights into the connections between the target and the graph, the target and the features, and the interplay between features and graph information. In some cases, an exact description of the model can be visualized through only a few figures. We also demonstrate how the interpretability of GNAN allows users to debug their model, a process that can be used for ensuring consistency with prior knowledge and avoiding biases and safety risks. In Section 5, we compare the performance of GNAN with other GNN architectures. This comparison underscores that sacrificing performance for intelligibility is not necessary, as the performance of GNAN is comparable to that of commonly used black-box GNNs.

The main contributions of this work are:

1. The extension of Generalized Additive Models (GAMs) to graph data.

2. The introduction of a fully interpretable-by-design model for graph prediction tasks, demonstrating that its explanations provide both global and local insights, through visualizations of the model itself, and include debugging capabilities.

3. The demonstration that GNAN achieves good performance on common real-world graph datasets, despite its limited capacity. This observation supports previous findings that some real-world graph problems are simple and do not require the capacity of other GNNs.

Thus we argue that GNAN is suitable for high-stakes applications due to its interpretability and performance.

## 2 Related work

**Generalized Additive Models** Generalized Additive Models (GAMs) are a class of statistical models that build upon generalized linear models by incorporating non-linear functions for each variable while maintaining additivity [19, 20, 21]. Essentially, GAMs model the expected value of the target variable as a sum of univariate functions of the features. Formally, in GAMs, a prediction

for an input $\mathbf{x}$ is computed by $\sigma \left( \sum f_k \left( \mathbf{x}_k \right) \right)$ where $\sigma$ is a predefined activation function, such as the sigmoid[2], and the $f_k$'s are shape functions learned during the training process. This approach extends generalized linear models, in which predictions are computed by $\sigma \left( \sum \mathbf{w}_k \mathbf{x}_k \right)$ where $w$ is a learned weight vector.

GAMs are more expressive than generalized linear models while remaining interpretable, as the effect of each predictor is modeled separately. For example, they can capture non-monotone effects of features, which generalized linear models cannot achieve without feature engineering. Traditionally, GAMs utilize splines or other smooth shape functions to model the non-linear relationships between each feature and the target variable. However, other methods, such as trees, have been proposed to fit the shape functions [24]. Recently, Agarwal et al. [26] suggested using neural networks to learn the shape functions. This approach combines the representational power of deep learning with the interpretability of additive models.

**Graph Neural Networks**   Graph Neural Networks (GNNs) [27, 28, 29, 30] have emerged as the leading approach for learning over graph data. The fundamental idea behind GNNs is to use neural-networks that combine node features with graph-structure. A commonly used family of GNNs is message-passing GNNs, where the representations of nodes are updated in iterations through neighborhood aggregations. This aggregation is done, for example, through a convolution-like operation or an attention mechanism. [31, 32, 33]

Various non-message-passing approaches have been explored to disentangle the node features from the graph structure. Such approaches were shown to enhance performance across diverse applications [34, 35, 36]. Disentanglement can also reduce overfitting, as popular GNNs which do entangle features and graph-structure, were shown to have the tendency to overfit non-informative graph information [37] GNAN uses these concepts in order to achieve a model that is both high-performing and fully interpretable.

There are different prediction tasks on graphs [38]. In *graph tasks*, the goal is to predict properties of entire graphs. For example, a graph could represent a molecule, and the goal would be to predict its toxicity level. In *node tasks*, the goal is to predict a property of a node (vertex) within a graph. An example of a node task is predicting whether a user in a social network is a human or a bot. In *link prediction tasks*, the goal is to determine whether there is an edge between two nodes of a graph. In this work, we focus on graph tasks and node tasks. Although link prediction tasks are not within the scope of this work, it is possible to view these problems as node tasks on the dual line graph [39].

**GNNs explanations**   The inherent complexity of graph-structured data poses unique challenges for explainability. Most approaches for explaining black-box GNNs focus on providing a sub-graph or a similar structure that can explain a certain example. This is done either as a post-hoc explanation for GNNs [40, 41, 42] or by adjusting the data a priory [43, 44, 45]. For example, the method suggested in Ying et al. [41] identifies both important subgraph structures and node features influencing the GNN's predictions by maximizing the mutual information between the prediction and the distribution of possible subgraph structures and node features. Yin et al. [43] suggested a structural pattern learning module that is learning through pre-training. GNAN, on the contrary to these methods, does not aim to provide an explanation through a proxy object like a subgraph, nor does it require modification to the data, or the training process. Instead, GNAN is a interpretable by design, and its exact description can be visualized through its learned shape functions. In particular, the exact relation between the target, the features, and the graph can be visualized and conveyed to users.

## 3   Graph Neural Additive Networks

In this section, we introduce the Graph Neural Additive Networks (GNAN). We begin by defining some essential notation. A graph $G$ has a set of $N$ vertices, where each vertex is associated with a $d$-dimensional feature vector. Specifically, $\mathbf{x}_i \in \mathbb{R}^d$ represents the feature vector of the $i$'th node in $G$. We define the distance $\text{dist}\,(j, i)$ between node $j$ and node $i$ within the graph $G$ as the number of edges in the shortest path from $j$ to $i$. This definition implies that the distance from a node to itself is zero. In cases where no path exists from $j$ to $i$, we set $\text{dist}\,(j, i) = \infty$. For enhanced readability, we denote vectors in boldface, and an entry $k$ of a vector $\mathbf{x}$ is denoted by $[\mathbf{x}]_k$. We begin

---

[2]In the context of Generalized Linear Models (GLMs) $\sigma$ can be though of as the inverse of the link-function.

by describing GNAN for applications such as binary classification and regression where the model output is one-dimensional. At the end of this section, we discuss extensions to scenarios such as multi-class classification, where the model output is multi-dimensional.

GNAN generates a representation $\mathbf{h}_i \in \mathbb{R}^d$ for each node $i$ by learning a distance function $\rho(x; \theta) : \mathbb{R} \to \mathbb{R}$ and a set of feature shape functions $\{f_k\}_{k=1}^d$, $f_k(x; \theta_k) : \mathbb{R} \to \mathbb{R}$. Each of these functions is a neural network, and is optimized through back-propagation. For brevity, we omit the parameterization $\theta$ and $\theta_k$ for the remainder of this section. In GNAN, the $k$'th entry of the representation $\mathbf{h}_i$ for the $i$'th node is defined as follows:

$$[\mathbf{h}_i]_k = \sum_{j=1}^N \frac{1}{\#\mathrm{dist}_i(j, i)} \cdot \rho\left(\frac{1}{1 + \mathrm{dist}(j, i)}\right) \cdot f_k\left([\mathbf{x}_j]_k\right)$$

where $\#\mathrm{dist}_i(j, i)$ represents the number of nodes at distance $\mathrm{dist}(j, i)$ from node $i$. The underlying rationale for this definition is as follows: each node's $k$'th feature is transformed by a shape function $f_k$, independently from other features. The effect the $k$'th feature value of node $j$ has on node $i$'s representation is influenced by their distance. Specifically, if $\mathrm{dist}(j, i) = l$, then the cumulative influence of all nodes at distance $l$ from node $i$ is captured by $\rho(1/(1+l))$. This is achieved by the normalization term $1/\#\mathrm{dist}_i(j,i)$. Here, $\rho$'s argument $1/(1+l)$ scales the distance such that a distance of $0$ (the self-distance of a node) is mapped to $1$, and an infinite distance, which implies no path exists, is scaled to $0$. Thus, $\rho$ spans the interval $[0, 1]$.

The representation of each node is dependent on the entire graph, yet the function $\rho$ enables weighting the influence from nodes, based on their distance. This enables, for example, diminishing the impact of distant nodes, or close neighbors. For each node $i$, the weighted sum of the transformed feature vectors of all other nodes is computed, with weights assigned according to their distance from $i$. This weighted sum is computed after the shape functions are applied to the distances and the features of each node.

Given the node representations, both node prediction tasks and graph prediction tasks can be implemented. To predict for the $i$'th node, the computation is as follows:

$$\sigma\left(\sum_{k=1}^d [\mathbf{h}_i]_k\right) \quad,$$

where the entry-wise sum of the representation vector $\mathbf{h}_i$ is computed and subsequently processed using an activation function such as the sigmoid for classification and the identity for regression. For a prediction over the entire graph, the collective node representation is computed via sum-pooling:

$$\mathbf{h} = \sum_{i=1}^N \mathbf{h}_i \quad.$$

Following this, the entry-wise sum of the graph representation $\mathbf{h}$ is computed and also processed using the activation function:

$$\sigma\left(\sum_{k=1}^d [\mathbf{h}]_k\right) \quad. \tag{1}$$

Once the model is trained, it can be fully described using its univariate functions $\rho$ and $\{f_k\}_{k=1}^d$.

From the definitions provided above, it follows that the entire model can be represented with just a few figures, thus providing global interpretability. For local explanations, it is feasible to examine the contribution of each feature and each node to the predictions. The following representation convey the influence of each node to the $k$'ts feature:

$$[\mathbf{h}]_k = \sum_{i=1}^N [\mathbf{h}_i]_k = \sum_{j=1}^N f_k\left([\mathbf{x}_j]_k\right) \sum_{i=1}^N \frac{1}{\#\mathrm{dist}_i(j, i)} \cdot \rho\left(\frac{1}{1 + \mathrm{dist}(j, i)}\right) \quad.$$

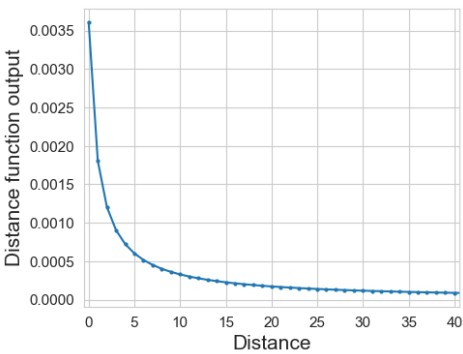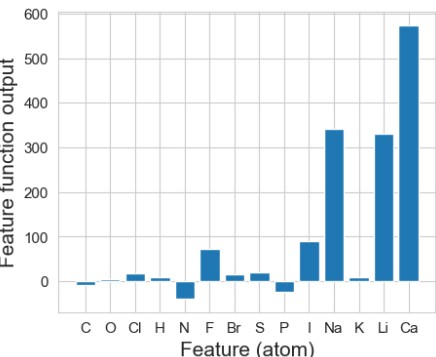

Figure 1: Visualization of the distance and feature functions, learned on Mutagenicity. As the features are binary, the feature functions are evaluated only on the value 1. These plots provide an exact description of the functions' signal processing and a global explanation of how the model uses the distances and features.

Here $[\mathbf{h}_i]_k$ contains the influence of the $k$'th feature in node $i$ on the prediction. However, from the definition of $[\mathbf{h}_i]_k$ we see that it serves as a mediator for influences of all other nodes. Therefore, the influence of node $j$ on feature $k$ in the final graph representation can be obtained by:

$$f_k\left([\mathbf{x}_j]_k\right) \sum_{i=1}^{N} \frac{1}{\#\text{dist}_i(j,i)} \cdot \rho\left(\frac{1}{1+\text{dist}(j,i)}\right) \quad . \tag{2}$$

We can also extract the total contribution of each node $i$ to the prediction, by summing the contribution of the nodes over the features, as done in the input to $\sigma$ in Equation 1:

$$\sum_{k=1}^{d}[\mathbf{h}]_k = \sum_{i=1}^{N}\sum_{j=1}^{N} \frac{1}{\#\text{dist}_i(j,i)} \cdot \rho\left(\frac{1}{1+\text{dist}(j,i)}\right) \sum_{k=1}^{d} f_k\left([\mathbf{x}_j]_k\right) \quad . $$

Therefore, the total contribution of node $i$ to the prediction is

$$\sum_{j=1}^{N} \frac{1}{\#\text{dist}_i(j,i)} \cdot \rho\left(\frac{1}{1+\text{dist}(j,i)}\right) \sum_{k=1}^{d} f_k\left([\mathbf{x}_j]_k\right) \quad . \tag{3}$$

Overall, the model facilitates a detailed understanding of local behavior from multiple perspectives.

The functions $\rho$ and $\{f_k\}_{k=1}^{d}$ may be implemented using a variety of neural network architectures. In our experiments, we employed multi-layer perceptrons (MLPs) with ReLU activations to implement these functions. Nonetheless, other alternatives are viable, such as employing learning splines for activations to achieve smoother shape functions [46]. Additionally, it is feasible to develop a separate distance network for each feature to enhance the model's capacity. Specifically, rather than utilizing a single function $\rho$, one can train a distinct function $\rho_k$ for each feature $k$, which weights the contribution of each feature based on its node's distance. For graph-level tasks, additional feature networks may be integrated prior to aggregating the graph's representation vector, akin to a readout layer in GNNs. These extensions, along with a discussion on a tensor representation of the computation that facilitates efficient GPU utilization, are further elaborated in the Appendix.

For multiclass classification involving $C$ classes, we configure the final layers of the feature shape functions $f_k(x;\theta_k) : \mathbb{R} \to \mathbb{R}^{C\times 1}$ and the distance function $\rho(x;\theta) : \mathbb{R} \to \mathbb{R}^{C\times 1}$ to accommodate the required dimensionality. The transformed feature vectors and the distance metrics are combined using an element-wise multiplication denoted by $\odot$, as follows:

$$[\mathbf{h}_i]_k = \sum_{j=1}^{N} \frac{1}{\#\text{dist}_i(j,i)} \cdot \rho\left(\frac{1}{1+\text{dist}(j,i)}\right) \odot f_k([\mathbf{x}_j]_k) \quad . $$

For prediction purposes, the sum operator is applied independently across the dimensions corresponding to each class, and a softmax is employed as the activation function.

# 4   Inteligibility

In this section, we demonstrate the intelligibility of GNAN through visualizations. Each GNAN model is characterized by the univariate learned shape functions $\rho$ and $\{f_k\}_{k=1}^d$, and can thus be depicted as a set of illustrative figures. Below, we present examples of such figures and explain their utility in generating insights. Our focus in this section is on global interpretability, as local interpretability can utilize analogous ways. We showcase GNAN's application on two datasets, with additional examples detailed in the Appendix.

Our initial examples focus on the task of detecting mutation-causing molecules using the Mutagenicity dataset [47]. In this task, molecules are modeled as graphs where nodes correspond to atoms and edges to connections between these atoms. Each atom type is represented by a 14-dimensional one-hot encoding. A GNAN model trained on this dataset is illustrated in Figure 1. On the left, the function $\rho$ is presented, demonstrating how distance impacts prediction, with a clear diminishing influence of more distant atoms. On the right, the shape functions for the features are displayed. Given that the features are binary, each shape function manifests only two values: one when the feature is $0$ (indicating that the atom is not of the specified type), and another when the feature is $1$ (indicating that the atom is of the specified type). Defining $b = \sum_k f_k(0)$ as the bias term allows us to set $f_k(0) = 0$ for each $k$, thereby enabling the plotting of only $f_k(1)$. The graphical representation reveals that atoms such as Ca, Na, and Li are predicted to correlate with an increased mutagenicity effect, whereas N and P atoms are predicted to be associated with a slight protective effect.

It is essential to emphasize that Figure 1 displays the entire model comprehensively. This means that combined with the value of the bias term, which is $-5.6672$ in this case, every crucial detail needed to understand and utilize this model for predictions is contained within this single figure. This stands in stark contrast to methods like feature importance, which offer a limited perspective on models. While the figure provides complete information about the model, presenting additional views can sometimes be helpful.

Figure 2 showcases the cross product of the shape functions and the distance function as a heatmap. Each cell $(k, l)$ in the heatmap represents the value $\rho\left(1/(1+l)\right) \cdot f_k(1)$. This figure illustrates the interplay the model has learned between the graph's structure and the attributes of its nodes. As the task involves binary classification, positive values in the heatmap contribute to classifying a molecule as mutagenic, whereas negative values indicate non-mutagenic properties.

This heatmap illustrates how atoms at specific distances influence the final outcome. For instance, it shows that the model has learned that the presence of a Ca atom (cell (Ca, 0)) or its proximity (cell (Ca, 1)) contributes to mutagenicity. The visualizations can also be used for debugging purposes. This can be crucial, for example, to ensure that the model is free from biases or to identify any discrepancies with existing scientific knowledge. If it is already known that Ca atoms actually have a negative effect on mutagenicity, users could identify and correct this misalignment in the model's learning. Additionally, this detailed understanding allows users to select models that not only achieve high accuracy on the given samples but also align with prior knowledge, optimizing both performance and reliability.

Interpreting multiclass prediction tasks poses significant challenges, as noted by Zhang et al. [6]. In this context, we showcase the interpretability of GNAN using the PubMed dataset [48]. This dataset comprises 19,717 scientific publications related to diabetes archived on PubMed and categorized into three distinct classes (type-1 diabetes, type-2 diabetes, and gestational diabetes). The dataset's citation network includes 44,338 links. Each publication, represented as a node, is characterized by a TF/IDF weighted word vector derived from a dictionary containing 500 unique words.

As there are three classes, we trained the GNAN model such that the output of the distance and feature functions are of dimension three. In this setting it is interesting to compare the three functions, corresponding to the three classes and therefore we draw them on the same figure [6]. Figure 3 shows that the model uses only the local neighborhood of each node, and as nodes become more distanced, the information between them is less used. We also observe a difference between the classes; while for type 2 diabetes, the longer the distance, the less their information is used (converges to 0), for type 1 and gestational diabetes, nodes of long-distance have a negative effect.

In Figure 4, we display the feature shape functions for nine selected features, demonstrating GNAN's capability to learn complex, non-monotone functions such as those seen in the 'diet' and 'hepat'

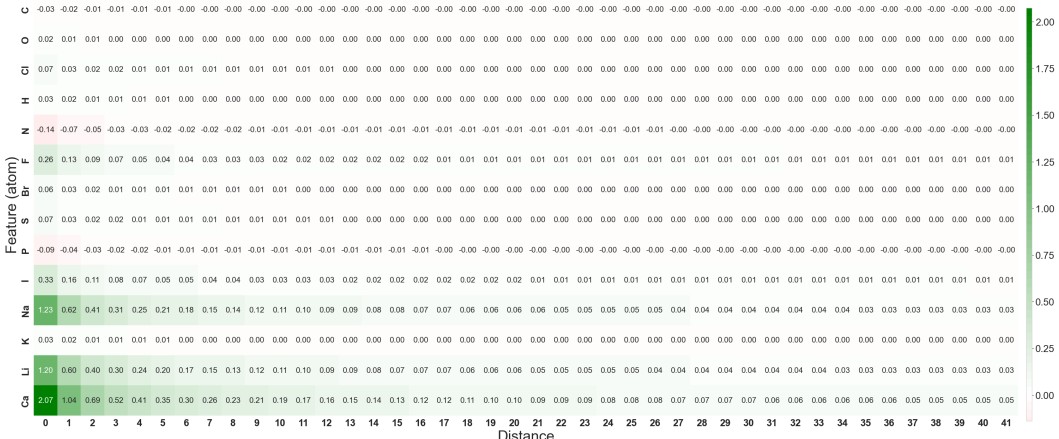

Figure 2: Visualization of products of the outputs of the distance function and the feature functions, trained on Mutagenicity. Each cell shows the exact contribution, positive or negative, of features at a certain distance to the prediction. Positive values (green) contribute to classifying a molecule as mutagenic, and negative values (red) contribute to classifying a molecule as non-mutagenic.

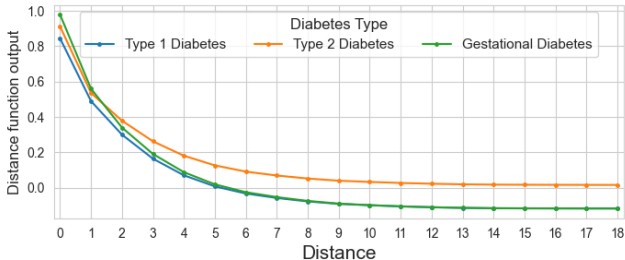

Figure 3: Visualization of the distance shape function learned on the PubMed dataset. As the output of the function is of dimension three, we plot it as three shape functions, one for each class. We plot them on the same figure to compare them. The shape functions show that the model uses only the local neighborhood of each node. It also shows a difference between the classes; while for type 2 diabetes, the longer the distance, the less their information is used (converges to 0), for type 1 and gestational diabetes, nodes of long-distance have a negative effect.

features. Observing these shape functions across the three classes simultaneously allows for an understanding of how different feature values are utilized by the model to distinguish among the classes. For instance, the shape function for the 'insulin' feature reveals that the absence of this word in a document (i.e., feature values close to zero) does not significantly indicate the document's class. However, as the frequency of 'insulin' increases within the document, its impact on the prediction becomes more pronounced, though this effect varies distinctly between type 1 & 2 diabetes and gestational diabetes.

To visualize the contribution of a feature value at a specific distance, we employ a heatmap for each class, evaluating the products between the outputs of the feature function over the input range ($[0, 1]$) and the output of the corresponding distance function. Figure 5 exemplifies this visualization technique with the 'children' feature. It is insightful to observe that the presence of the word 'children' influences the predictions differently across the diabetes types. The model has learned that papers concerning type 1 diabetes seldom mention 'children', nor do related papers. In contrast, the term frequently appears in the context of gestational diabetes.

It is possible to construct confidence intervals for GAMs using the bootstrap method[49, 50]. We present such example with additional visualization examples in the appendix.

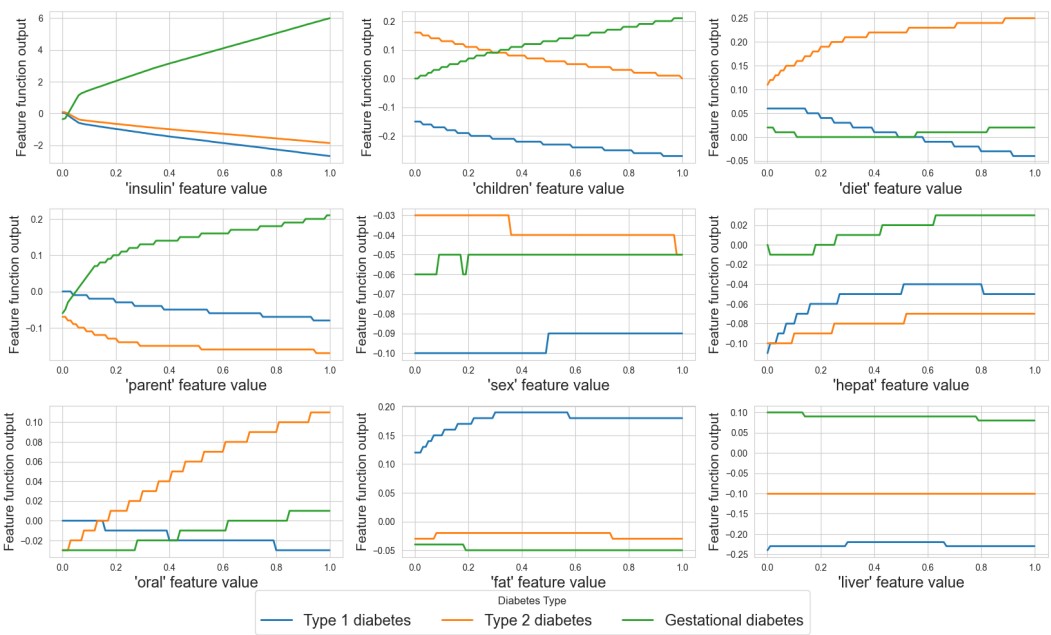

Figure 4: Visualization of nine features' shape functions, learned over the PubMed dataset.

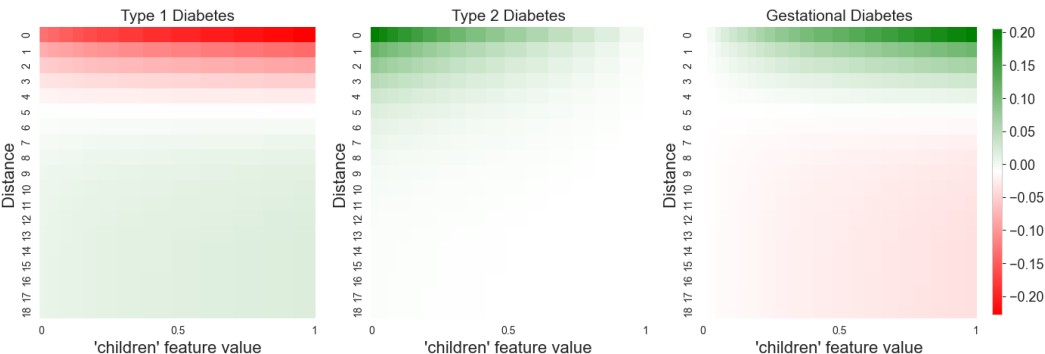

Figure 5: Visualization of the products between the outputs of the 'children' feature function over the input range $[0, 1]$ and the outputs of the distance function, learned over the PubMed dataset.

## 4.1 Local Explanations

Up to this point, we have demonstrated how to visualize GNAN for providing global explanations. These explanations offer a comprehensive visualization of the entire model. Now, we shift our focus to local explanations, i.e., those relevant to particular examples of interest. To illustrate this, we employ the same Mutagenicity model that was visualized globally to explain specific samples within the data. Using Equation 3, we compute the importance of each node and visualize two molecules from the dataset, where the area of each node corresponds to its importance.

Figure 6 presents two such examples. In Figure 6(a) the carbon (red) atoms play the most significant role, and a carbon ring (red cycle) is highlighted. In Figure 6(b), a group of $NO_2$ (grey and green subgraph) is shown to be relatively important for predicting the molecule as mutagenic. Both carbon rings and $NO_2$ groups are well-known for their mutagenic effects [51], making them frequently discussed in the literature on explainable GNNs [39, 41].

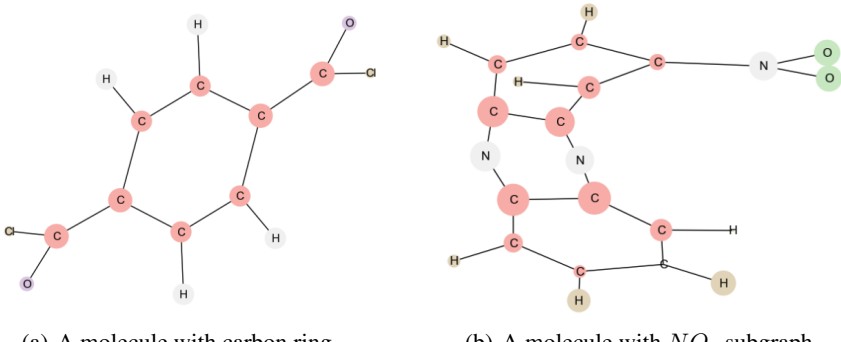

|(a) A molecule with carbon ring | (b) A molecule with $NO_2$ subgraph|

Figure 6: Local explanations of two molecules from the Mutagenicity datasets, through visualizations of the molecules. The area of each atom corresponds to the node importance according to Equation 3.

## 5    Empirical evaluation

In this section, we evaluate GNAN on real-world graph and node labeling tasks, including large-scale, long-range, and heterophily datasets.[3]. We compare GNAN to multiple commonly used black-box GNNs including GraphConv [52], GraphSAGE [30], Graph Isomorphism Network (GIN) [33], the expressive version of the Graph Attention Network (GATv2) [29, 53], the Graph Transformer (GTransformer) [54]. We also evaluate the FSGNN model, which disentangles the node features from the graph structure [35]. The information on the hyper-parameters tuned for each baseline can be found in the Appendix. We used the following common benchmarks:

**Node labeling tasks** *Cora, Citeseer, PubMed, ogb-arxiv* [55, 56] are paper citation networks where the goal is to classify papers into one of several topics. The ogb-arxiv dataset is a large-scale network. *Cornell [57] & Tolokers [58]* are heterophilious datasets. Cornell is a web-link network with the task of classifying nodes into one of five categories. Tolokers dataset is based on data from the Toloka crowdsourcing platform. The nodes represent tolokers (workers) who have participated in at least one of 13 selected projects. An edge connects two tolokers if they have worked on the same task. The goal is to predict which tolokers have been banned in one of the projects. Node features are based on the worker's profile information and task performance statistics.

**Graph labeling tasks** *NCI1, Proteins, Mutagen & PTC*  [59] are datasets of chemical compounds. In each dataset, the goal is to classify compounds according to some property of interest. Thr $\mu$ ,$\alpha$ ,$\alpha_{HOMO}$ [60] datasets are long-range molecular property prediction regression tasks, over the large-scale QM9 molecular dataset.
Additional data information, including the data statistics, can be found in the Appendix.

**Protocol**    For all tasks, we used existing splits, protocols, and metrics, as commonly used in the literature for each dataset. The complete protocols for each dataset are given in detail in the Appendix. The metrics we report are: For Cornell, Cora, Citeseer, PubMed, ogb-arxiv, Mutagenicity, PTC, NCI, and Proteins, we report accuracy. For $\mu$, $\alpha$ and $\alpha_{HOMO}$ we report MAE. For Tolokers we report ROC-AUC. For the node labeling tasks, we used the pre-defined splits in the data and followed the common protocols for each dataset. The results are an average of the test set using 5 or 10 random seeds. For the Proteins and NCI1 tasks, we followed the splits and the nested-cross-validation protocol from [61]. The final reported result on these datasets is an average of 30 runs (10-folds and 3 random seeds). For NCI1 and PTC we followed the splits and protocol from [39] and report the average accuracy and std of a 10-fold nested cross-validation.

**Results**    The results are presented in Table 1. GNAN performed as the best or second-best model in 9 out of the 13 tasks we evaluated. In GNAN, each node gathers information from all others, ensuring complete information flow, while the $\rho$ function modulates influence based on distance.

---

[3]The implementation can be found at `https://github.com/mayabechlerspeicher/Graph-Neural-Additive-Networks---GNAN`

Table 1: Evaluation of GNAN on node (top) and graph (bottom) tasks. The best and second-best models are marked in cyan and violet colors, respectively. We report accuracy and std for all tasks, except for the Tolokers dataset where we report ROC-AUC and std, and the $\mu$, $\alpha$, $\alpha_{HOMO}$ datasets where we report MAE and std.

| Model | Cornell | Tolokers | Cora | Citeseer | PubMed | ogb-arxiv |
|---|---|---|---|---|---|---|
| GraphConv | 65.9 ± 0.5 | 83.5 ± 0.7 | 81.3 ± 1.1 | 75.9 ± 2.0 | 85.9 ± 0.5 | 72.4 ± 0.1 |
| GraphSAGE | 75.9 ± 5.0 | 82.4 ± 0.4 | 81.4 ± 0.7 | 76.4 ± 0.8 | 88.4 ± 0.4 | 71.7 ± 0.2 |
| GIN | 69.0 ± 1.3 | 81.0 ± 0.4 | 80.0 ± 1.2 | 77.1 ± 1.9 | 85.3 ± 0.9 | 73.8 ± 1.4 |
| GATv2 | 72.5 ± 0.7 | 83.8 ± 1.1 | 83.1 ± 0.9 | 73.9 ± 1.5 | 84.4 ± 0.5 | 74.0 ± 2.1 |
| GTransformer | 70.5 ± 1.7 | 83.3 ± 0.9 | 80.7 ± 0.5 | 76.0 ± 0.9 | 85.3 ± 1.6 | 73.1 ± 0.2 |
| FSGNN | 86.0 ± 4.1 | 83.1 ± 0.6 | 83.0 ± 1.3 | 76.2 ± 1.3 | 85.0 ± 1.3 | 72.9 ± 1.7 |
| GNAN | 85.7 ± 4.8 | 84.5 ± 0.9 | 81.1 ± 1.5 | 75.8 ± 0.6 | 86.9 ± 1.2 | 74.1 ± 1.5 |

| Model | $\mu$ | $\alpha$ | $\alpha_{HOMO}$ | Proteins | Mutagen | PTC | NCI1 |
|---|---|---|---|---|---|---|---|
| GraphConv | 2.91 ± 0.1 | 4.37 ± 0.5 | 1.46 ± 0.1 | 73.1 ± 1.6 | 64.3 ± 1.7 | 63.9 ± 5.0 | 76.5 ± 1.2 |
| GraphSAGE | 3.55 ± 0.2 | 4.51 ± 0.7 | 1.44 ± 0.2 | 73.0 ± 4.5 | 64.1 ± 0.3 | 67.1±12.6 | 76.0 ± 1.8 |
| GIN | 2.60 ± 0.1 | 4.67 ± 0.5 | 1.42 ± 0.1 | 73.3 ± 4.0 | 69.4 ± 1.2 | 55.6±11.1 | 80.0 ± 1.4 |
| GATv2 | 2.72 ± 0.1 | 4.39 ± 0.6 | 1.41 ± 0.1 | 73.5 ± 2.8 | 72.0 ± 0.9 | 59.5 ± 2.1 | 80.4 ± 1.6 |
| GTransformer | 2.90 ± 0.3 | 4.30 ± 0.5 | 1.41 ± 0.2 | 73.9 ± 1.5 | 73.1 ± 0.9 | 55.9 ± 3.5 | 80.5 ± 1.1 |
| FSGNN | 3.57 ± 0.3 | 4.50 ± 0.4 | 1.44 ± 0.3 | 72.9 ± 2.1 | 66.9 ± 1.5 | 60.3 ± 7.2 | 79.7 ± 1.1 |
| GNAN | 2.55 ± 0.1 | 4.28 ± 0.9 | 1.40 ± 0.1 | 73.2 ± 3.1 | 72.2 ± 1.0 | 64.9 ± 7.1 | 76.9 ± 1.2 |

Consequently, GNAN avoids the computational bottlenecks encountered by some message-passing GNNs [25]. Particularly in the long-range tasks $\mu$, $\alpha$, and $\alpha_{HOMO}$, GNAN outperformed all other evaluated baselines, aligning with findings by Alon and Yahav [25] that emphasize the benefits of capturing long-range information. While intelligibility sometimes comes at the cost of accuracy, our findings suggest that enhancing intelligibility does not necessarily result in significant accuracy loss. It may appear surprising that GNAN, despite its limited capacity, matches the accuracy of more expressive GNNs. However, prior research indicates that even limited-capacity GNNs, such as linear GNNs, can achieve high accuracy on various real-world datasets [62, 61, 63], suggesting that some real-world graph problems are simpler than anticipated. Our results corroborate these observations.

## 6 Conclusion

In this work, we introduced the Graph Neural Additive Network (GNAN), a novel extension of the interpretable class of Generalized Additive Models, to accommodate graph data. GNAN is inherently interpretable, and provides both global and local explanations directly from its architecture, eliminating the need for post-hoc interpretations. This direct interpretability enhances the transparency of the model and is particularly useful in high-stakes applications where understanding model decisions is crucial. Furthermore, GNAN demonstrates competitive performance with popular GNNs, showing that intelligibility does not necessarily entail a significant degradation in accuracy.

It is possible to enhance GNAN in several ways. To generate smooth shape functions, one could integrate techniques from the recently proposed Kolmogorov–Arnold Networks [46] or adaptive activations for graphs [64]. Increasing the capacity of GNAN is feasible by learning individual distance functions for each feature. Exploring reduced capacity is also intriguing, particularly in scenarios with many features, where it may be beneficial to employ regularization to limit the number of shape functions used. Additionally, applying these techniques to biological network datasets, such as protein interactions, could be a valuable tool to support scientific discoveries. These and other directions are left for future studies.

## Acknowledgements

This work was supported by the Tel Aviv University Center for AI and Data Science (TAD) and the Israeli Science Foundation grants 1186/18 and 1437/22.

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

# A Efficient GNAN implementation with tensor products

We formulate GNAN case of classification with $C$ classes. For regression, the exact formulation holds with $C = 1$. We use the same notation as in the main text, only now the output of the feature and distance function is of dimension $1 \times C$,

For the sake of notation, we assume that every tensor that its last dimension is of size $C$, is permuted to have the last dimension as its first dimension, without stating it explicitly. This is necessary to achieve a valid tensor multiplication.

We denote with $M$ the matrix of the transformed distances that is outputted by applying $\rho$, i.e., $M_{i,j} = \rho(\frac{1}{1+dist(j,i)})$, $M \in \mathbb{R}^{C \times N \times N}$.

We denote with $F$ the matrix of the transformed feature is outputted by applying the corresponding $f_k$ for feature $l$ of each node in the graph, i.e., $F_{ik} = f_k(x_l^i)$, $F \in \mathbb{R}^{C \times N \times d}$

Both for node and graph tasks, we first computes the matrix $M \cdot F \in \mathbb{R}^{C \times N \times d}$

The rest of the computation then depends on the task.

## A.1 Node Tasks

We aggregate the transformed features weighted by the transformed distances. This is done by summing over the rows of $M \otimes F$ :

$$\phi(i) = [M \otimes F \otimes \mathbb{1}_{C \times d \times 1}]_i = \sum_{k=1}^{d} \sum_{j=1}^{N} \rho(\frac{1}{1 + dist(j, i)}) \otimes f_k([\mathbf{x}_j]_k)$$

## A.2 Graph Tasks

For graph classification, we set $\rho$ and $f$ to output a scalar and aggregate the transformed features over the nodes, i.e., the row of $M \cdot F$, to form a fixed-size vector of size $d$.

$$\bar{\Phi}(G) = \mathbb{1}_{C \times 1 \times N} \otimes M \otimes F \in \mathbb{R}^{C \times 1 \times d}$$

Then, we can apply another *readout NAM [26]*:

$$\Phi(G) = \bar{F}(\bar{\Phi}(G))$$

Such that $\bar{F}$ is the transformed features using the function $\{\bar{f}_k\}_{k=1}^d$ such that $\bar{f}_k : \mathbb{R} \to \mathbb{R}^{1 \times C}$

We can also simply sum over the outputs. In that case, we will set $f$ and $m$ to output a vector of dimension $C$:

$$\Phi(G) = \sum_{i=1}^{N} \phi(i) = \mathbb{1}_{C \times 1 \times N} \otimes M \otimes F \otimes \mathbb{1}_{C \times d \times 1}$$

# B Extensions and ablations

In Section 3 we mentioned several possible extensions for GNAN. Here, we discuss them in detail.

**Readout layer for graph tasks** In graph tasks, after aggregating the node representations, it is possible to apply another transformation before aggregating over the entries of the graph representations. There may be many ways to do so, and we did not explore all of them. We did explore an application of another set of feature functions to each feature or the graph representation vector. This approach increases the capacity of the model in the cost of interpretability. This is because the set of addition feature functions should be plotted separately, and the product between the feature function and the distance functions does not affect the final output directly but rather through another feature function. Empirically, we observed this approach did not improve performance with respect to the performance reported in Section 5.

**Splines**  It is possible to learn splines for the activations in each feature network to achieve smoother shape functions [46]. We note that the Tolokers example presented in Section C shows that the learned feature shape function is smooth, although we use ReLU activations. Nonetheless, in other cases, such as in the PubMed example presented in the main paper, many of the learned feature functions are step functions. Therefore, it is likely that the model could benefit from spline activations, to smooth its shape functions.

**Normalization**  In GNAN we normalize the weight of nodes of distance $l$ with the number of nodes of distance $l$, so that the cumulative weight of nodes of distance $l$ will be $\rho(1/(1+l))$. We examined the effect of removing this normalization. We observed that without normalization, the loss scale is drastically larger. Therefore, more epochs are required to fit the data. As a result, for the fixed number of epochs we used in our experiments (1000), without normalization, the accuracy decreases.

## C    Additional explanation examples

In the main paper, we presented two examples of explanations over two datasets with different properties. In this section, we present additional explanations and examples we could not fit into the main text due to space limitations.

### C.1    Confidence Intervals

In our main paper, we discussed the construction of confidence intervals for Generalized Additive Models (GAMs) using the bootstrap method. Here, we provide a specific example using the Mutagenicity dataset.

We computed 95% confidence intervals by applying the bootstrap method with 200 resamples. This involved resampling the original dataset with replacement 200 times and calculating the statistic of interest for each resample. The resulting bootstrap estimates were sorted, and the 2.5th and 97.5th percentiles were taken as the lower and upper bounds of the confidence interval, respectively. Figure 7 presents the distance function with its confidence intervals.

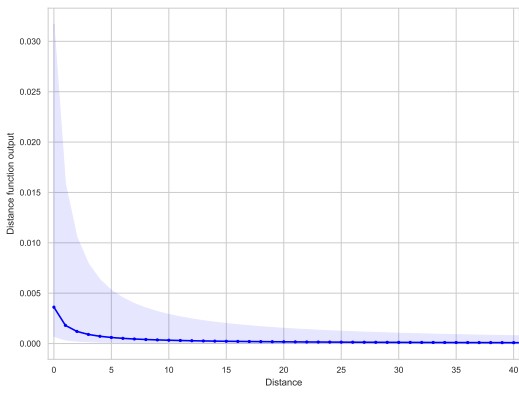

Figure 7

### C.2    Additional PubMed heatmaps

In the main text, we presented the heatmaps for the 'children' feature. Here we provide additional heatmaps for additional features: the 'fat' feature and the 'young' feature, as presented in Figures 8 and 9.

### C.3    Tolokers - Binary classification with binary and continuous features

The Tolokers dataset is based on data from the Toloka crowdsourcing platform. The nodes represent tolokers (workers) who have participated in at least one of 13 selected projects. An edge connects two tolokers if they have worked on the same task. The goal is to predict which tolokers have been

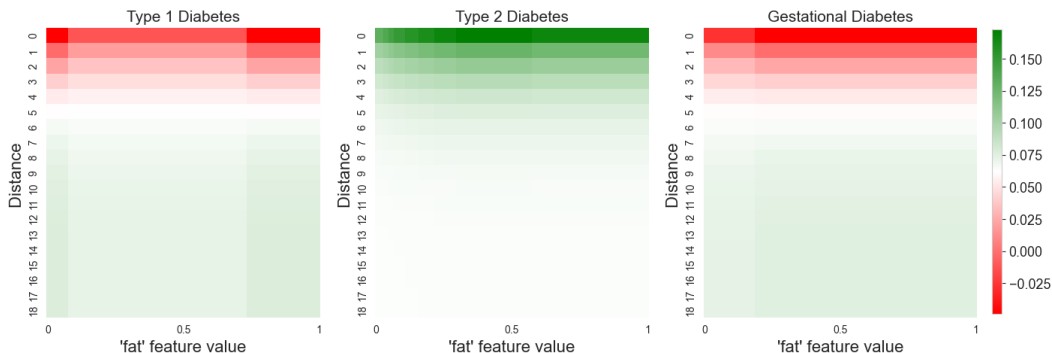

Figure 8: Visualization of the products between the outputs of the 'fat' feature function over the input range [0, 1] and the outputs of the distance function, learned over the PubMed dataset

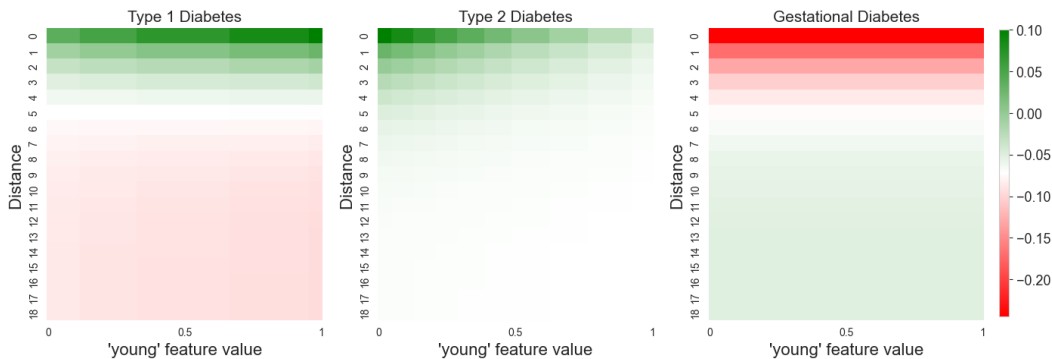

Figure 9: Visualization of the products between the outputs of the 'young' feature function over the input range [0, 1] and the outputs of the distance function, learned over the PubMed dataset

banned in one of the projects. Node features are based on the worker's profile information and task performance statistics. Each node in the graph is associated with nine features. There are 4 continuous features in the range [0, 1] and 5 are binary features. Figure 10 shows the shape functions of the features learned by GNAN. Figure 11 shows the distance shape function learned by GNAN. In Figures 13 and 12 we present the heatmaps of the cross product of the shape functions and the distance function.

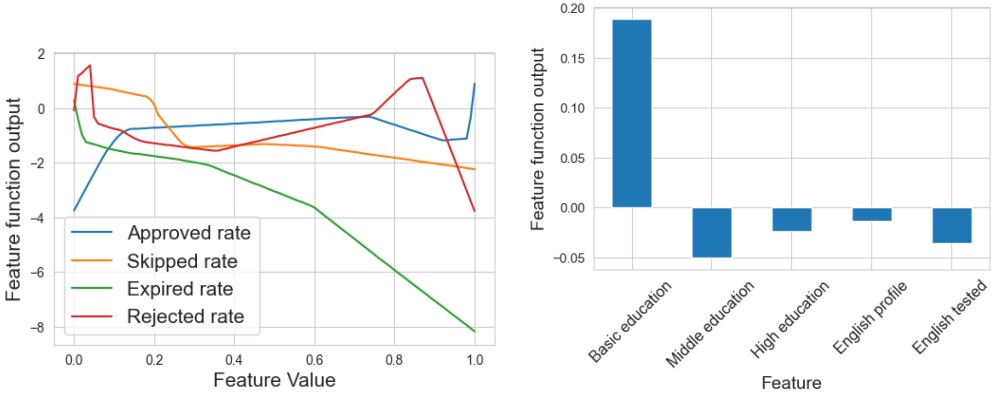

Figure 10: Visualization of the feature functions learned over the Tolokers dataset.

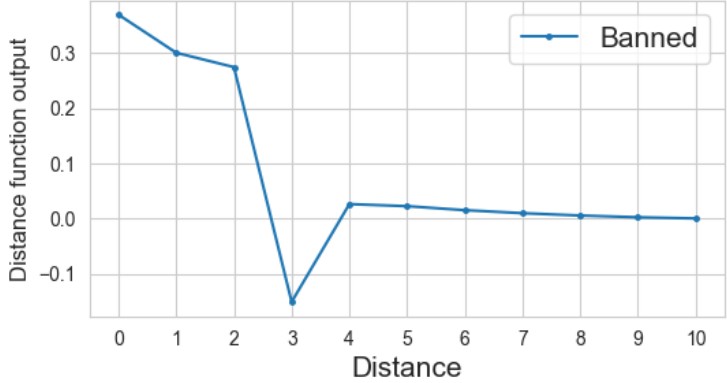

Figure 11: isualization of the distance shape function learned on the Tolokers dataset.

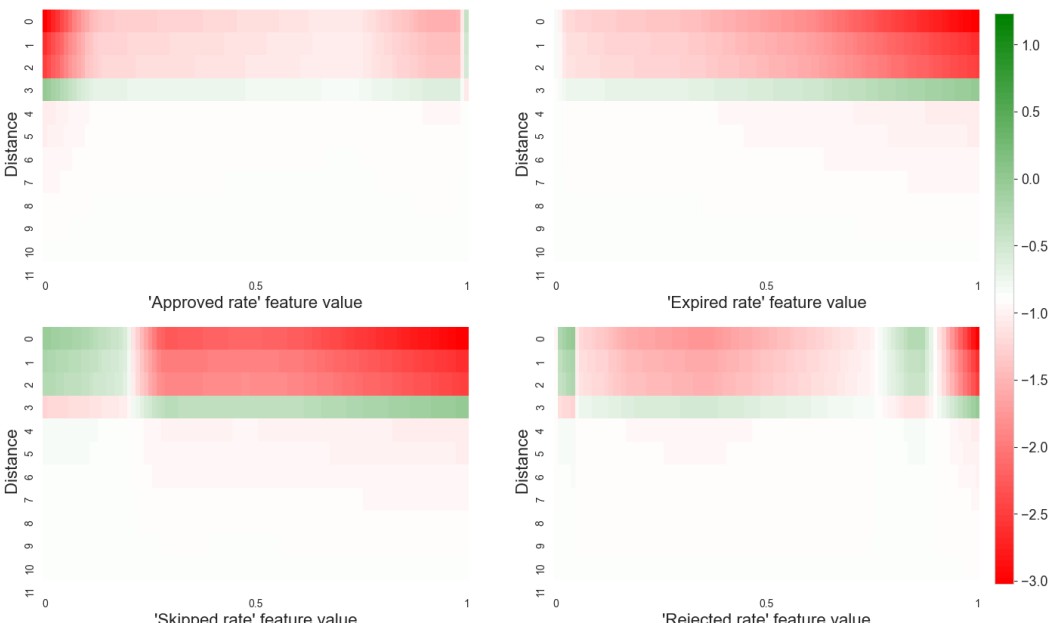

Figure 12: Visualization of the products between the outputs of the continuous feature functions over the input range $[0, 1]$ and the outputs of the distance function, learned over the Tolokers dataset.

## D   Additional experimental details

All our baselines are implemented using PyTorch [65] and PyTorch-Geometric [66].

### D.1   Dataset information

Here we provide additional information about the datasets used in Section 5. The data statistics are given in Table 2.

**Proteins** [59] is a dataset of chemical compounds consisting of 1113 graphs, respectively. The goal in the first two datasets is to predict whether a compound is an enzyme or not, and the goal in the last datasets is to classify the type of an enzyme among 6 classes.

**NCI1** [59] is a datasets consisting of 4110 graphs, representing chemical compounds. Vertices and edges represent atoms and the chemical bonds between them. The graphs are divided into two classes

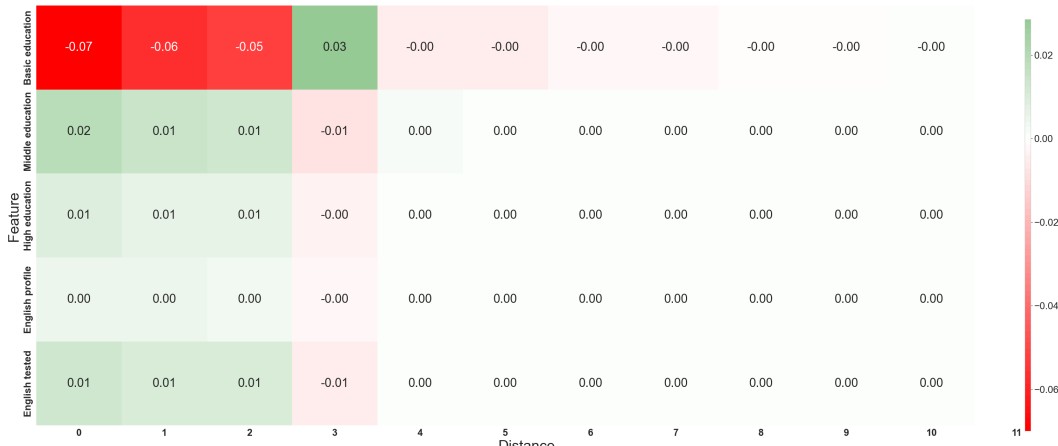

Figure 13: Visualization of products of the outputs of the distance function and the feature functions, trained on Tolokers. Each cell shows the exact contribution, positive or negative, of features at a certain distance to the prediction. Positive values (green) contribute to classifying a toloker as 'banned', and negative values (red) contribute to classifying a toloker as 'not banned'.

according to their ability to suppress or inhibit tumor growth.

**Mutagenicity** [59] is a dataset consisting of 4337 chemical compounds of drugs divided into two classes: mutagen and non-mutagen. A mutagen is a compound that changes genetic material such as DNA, and increases mutation frequency.

**PTC** [59] is a dataset consisting of 344 chemical compounds divided into two classes according to their carcinogenicity for rats.

**Cornell** [57] is a heterophilic webpage dataset collected from the computer science department at Cornell University. Nodes represent web pages, and edges are hyperlinks between them. The task is to classify the nodes into one of five categories.

Table 2: Statistics of the real-world datasets used in our evaluation.

| Dataset | # Graphs | Avg # Nodes | Avg # Edges | # Node Features | # Classes |
|---|---|---|---|---|---|
| Proteins [59] | 1,113 | 39.06 | 72.82 | 3 | 2 |
| NCI1 [59] | 4,110 | 29.87 | 32.3 | 37 | 2 |
| Mutagenicity [59] | 4,337 | 30.32 | 30.37 | 7 | 2 |
| PTC [59] | 344 | 14 | 14 | 19 | 2 |
| QM9 [60] | 130,831 | 18 | 37.3 | 11 | - |
| Cora [55] | 1 | 2,708 | 10,556 | 1,433 | 7 |
| Citeseer [55] | 1 | 3,327 | 9,104 | 3,703 | 6 |
| PubMed [55] | 1 | 19,717 | 88,648 | 500 | 3 |
| ogb-arxiv [56] | 1 | 169,343 | 1,166,243 | 128 | 40 |
| Cornell [57] | 1 | 183 | 295 | 1,703 | 5 |
| Tolokers [58] | 1 | 11758 | 519000 | 10 | 2 |

## D.2 Protocols

**ogb-arxive** The ogb-arxive datasets are large-scale datasets provided in the Open Graph Benchmark (OGB) paper [56] with pre-defined train and test splits and different metrics and protocols for each dataset. As common in the literature when evaluating OGB datasets, we followed its pre-defined

metric and protocol. The metric used is accuracy. We ran GNAN 10 times and reported the mean accuracy and std over the runs.

**Cornell**    For the Cornell dataset we used the splits and protocol from [57] and report the test accuracy averaged over 10 runs, using the best hyper-paremeters found on the validation set.

**Tolokers**    For the Tolokers dataset, we followed the protocol and pre-defined splits from [58, 67]. The reported result is an average of a 10-fold nested cross-validation.

**Core, Citeseer and PubMed**    Following [68, 30, 29], for the Core, Citeseer and Pubmed datasets we tuned the parameters on the Cora dataset using the pre-defined splits from [68]. For all these datasets we report the test accuracies averaged over 5 runs, using the parameters obtained from the best accuracy on the validation set of Cora.For Cora and Citeseer, we followed the schemes used in [69] where the used graph is the largest connected component, to increase the stability of the evaluated models.

**Proteins, NCI**    We used 10-fold nested cross validation with the splits and protocol of Errica et al. [61]. The final reported result on these datasets is an average of 30 runs (10-folds and 3 random seeds).

**Mutagenicity, PTC**    We use the splits and protocols from [39], and use a 10-fold nested cross-validation. The final reported test accuracies are averages over the 10 test sets of the outer 10 folds.

### D.3    Hyperparameters

All GNNs (excluded GNAN) use ReLU activations with $\{3, 5\}$ layers and 64 hidden channels. They were trained with Adam optimizer over 1000 epochs and early on the validation loss with a patient of 100 steps, eight Decay of $1e - 4$, learning rate in $\{1e - 3, 1e - 4\}$, dropout rate in $\{0, 0.5\}$, and a train batch size of 32.

In GNAN, all the feature and distance networks use ReLU activations with $\{3, 5\}$ layers and $\{64, 32\}$ hidden channels. They were trained with Adam optimizer over 1000 epochs weight decay of $0, 5e - 4$, learning rate in $\{1e - 2, 1e - 3\}$, dropout rate in $\{0, 0.6\}$.

### D.4    Compute resources

All experiments ran on an NVIDIA GeForce RTX 3090 GPU.

