# OpenReview forum: "The Intelligible and Effective Graph Neural Additive Network"
_NeurIPS.cc/2024/Conference — NeurIPS 2024 poster_

### Official Review · Reviewer_uKSz · 2024-06-25

**Soundness:** 3
**Presentation:** 3
**Contribution:** 2
**Rating:** 5
**Confidence:** 3

**Summary:**

This paper proposes additive Graph Neural Networks. A combination of interpretable neural additive models and Graph neural networks. Through this combination GNANs are interpretable and similarly to NAMs the feature effects are visualizable.

**Strengths:**

- The paper is overall well written
- The idea is simple yet intuitive

**Weaknesses:**

- If I am not mistaken the authors are not performing any hyperparamter optimization. Thus, the experimental results, while they average over a lot of folds/seeds are not as conclusive as it might seem
- The interpretability is not tested. The visualizations could be completely biased. There should at least be an ablation study with simulated feature effects.
- While the feature effects may be visualizable, what about intelligebility (e.g. [1])
- There has been quite a lot of research in the NAM area, for e.g. the architecture of the feature networks [2, 3, 4, 5] or distributional approaches [6]. None of these is mentioned and the chosen GNAN architecture with MLPs as feature nets is already a bit outdated.



Minor:
- the initial introduction of the GAM could be a bit better. You are missing an intercept, possible feature interactions and while you later define vectors to be bold do not adjust for that in this section.
- The appendix is very poorly written with a lot of errors, both grammatically and spelling.

[1] Luber, M., Thielmann, A., & Säfken, B. (2023). Structural neural additive models: Enhanced interpretable machine learning. arXiv preprint arXiv:2302.09275.

[2] Chang, C. H., Caruana, R., & Goldenberg, A. (2021). Node-gam: Neural generalized additive model for interpretable deep learning. arXiv preprint arXiv:2106.01613.

[3] Radenovic, F., Dubey, A., & Mahajan, D. (2022). Neural basis models for interpretability. Advances in Neural Information Processing Systems, 35, 8414-8426.

[4] Thielmann, A. F., Reuter, A., Kneib, T., Rügamer, D., & Säfken, B. Interpretable Additive Tabular Transformer Networks. Transactions on Machine Learning Research.

[5] Dubey, A., Radenovic, F., & Mahajan, D. (2022). Scalable interpretability via polynomials. Advances in neural information processing systems, 35, 36748-36761.

[6] Thielmann, A. F., Kruse, R. M., Kneib, T., & Säfken, B. (2024, April). Neural additive models for location scale and shape: A framework for interpretable neural regression beyond the mean. In International Conference on Artificial Intelligence and Statistics (pp. 1783-1791). PMLR.

**Questions:**

- How do you explain that GNANs, although they have the additivity constraint outperform fully connected networks?
- Why do you choose different architectures for GNANs and all comparison models? Also different learning rates, weight decay etc? If no hyperparamter tuning is performed, the architectures should at least be identical
- Why choose MLPs as feature networks when they have been outperformed by newer (and older) architectures?
- What are the model parameters compared to the benchmarks? Given that NAMs often have a multitude of fully connected networks, I would expect the same here?

**Limitations:**

yes

---

> ### Author Rebuttal · Authors · 2024-08-03
>
> We thank the reviewer for the thoughtful comments and encouraging remarks.
>
> ## Weakness 1:
> The reviewer mentions that no hyper-parameters were tuned. We did tune the hyper-parameters, and the grid of hyper-parameters is presented in Appendix D3 and referred to in line 252.
>
> ## Weakness 2:
> The reviewer suggests testing the interpretability of GNAN. GAMs are considered explainable models [1,2]. The explanations they provide make the model transparent in the sense that the way the model makes its predictions is humanly comprehensible. However, it is important to distinguish between explaining the model and explaining the underlying data, which often requires causal analysis [3]. \
> While the model may have some inductive biases, the visualization makes the model transparent to users, allowing them to scrutinize and adjust it if needed. Performing a usability study to check the interpretability of the solution is a great idea but is beyond the scope of this study.
>
>
> ## Weakness 3:
> The reviewer asks about the intelligibility of GNAN and refers to a paper that extends NAMs to a stronger neural architecture and proposes methods to extend the intelligibility of NAMs. We thank the reviewer for the suggestion showing that there is great potential for further research in this field. Regarding intelligibility, as GNAN is an instance of GAMs, it is intelligible in the same sense that GAMs are.
> ## Weakness 4:
> The reviewer mentions that MLPs are outdated with respect to the NAMs literature and referred to newer approaches. We thank the reviewer for these references. GNAN is the first extension of GAMs to graphs, and we decided to utilize the commonly used MLPs. The fact that there is potential for improving GNAN even further presents a great future research direction. \
> We demonstrated that even with MLPs, GNAN is on par with black-box GNNs while providing full interpretability. We will make sure to refer to these studies and the future research direction in our camera-ready version.
>
>
> ## Weakness 5 minors:
> The reviewer suggests improving the introduction of GNAN. We appreciate this comment and we will improve it in the camera-ready version.
>
> ## Weakness 6 minors:
> The reviewer suggests improving the appendix. We appreciate this comment and we will improve it in the camera-ready version.
>
>
> ## Question 1:
> The reviewer asks how GNAN outperforms fully connected networks despite its limited capacity. We would greatly appreciate it if the reviewer could clarify what they mean by “fully connected networks” in the context of graph networks. \
> We compare GNAN to methods that are not restricted in the same way GNAN is, including sparse graph transformers [4]. The empirical evidence reported in this paper shows that GNAN is mostly on par with these methods and sometimes even outperforms them.
>
> ## Question 2:
> The reviewer asks why different hyperparameters are selected for different models if no hyperparameter tuning was performed. We wish to reiterate our response to Weakness 1: we did tune hyperparameters in a standard fashion as described in Appendix D3.
>
> ## Question 3:
> The reviewer suggests replacing MLPs with other newer networks. Please see our response to Weakness 4.
>
> ## Question 4:
> The reviewer asks what the parameters in GNAN are. If the reviewer is referring to hyperparameters, these are listed in Appendix D3.
>
>
> [1]Interpretability, then what? editing machine learning models to reflect human knowledge and values, Chang et al, 2021. \
> [2]How interpretable and trustworthy are gams, Wang et al, 2021. \
> [3] True to the model or true to the data? Chen, H., Janizek, J. D., Lundberg, S., & Lee, S. I., 2020. \
> [4] Masked label prediction: Unified message passing model for semi-supervised classification, Shi et al, IJCAI 2021.

---

> > ### Comment · Area_Chair_wJum · 2024-08-07
> >
> > Thank you for the response. Dear reviewer uKSz, could you check whether the authors addressed your concerns? Thank you!

---

> > ### Comment · Reviewer_uKSz · 2024-08-09
> > **Rebuttal Answer**
> >
> > Dear Authors,
> >
> > Thank you for your response. However, my reservations remain.
> >
> > > We did tune the hyper-parameters, and the grid of hyper-parameters is presented in Appendix D3 and referred to in line 252.
> >
> > Thank you for clarifying. I would suggest to provide more details on the tuning process in the final version. For example, how many trials were conducted for each model?
> >
> > > Regarding intelligibility, as GNAN is an instance of GAMs, it is intelligible in the same sense that GAMs are.
> >
> > I strongly disagree. GNANs are as intelligible as NAMs, meaning they are visualizable. However, GAMs offer intelligibility beyond mere visualization through hypothesis testing.
> >
> > > While the model may have some inductive biases, the visualization makes the model transparent to users, allowing them to scrutinize and adjust it if needed. Performing a usability study to check the interpretability of the solution is a great idea but is beyond the scope of this study.
> >
> > I see your point. However, in its current form, the paper simply displays some graphics of GNANs’ predictions. If these are not tested against the true underlying data (which I still believe should be within the scope of this paper), they should at least be compared to another explainable model or post-hoc explainable approaches.
> >
> > > We compare GNAN to methods that are not restricted in the same way GNAN is, including sparse graph transformers [4]. The empirical evidence reported in this paper shows that GNAN is mostly on par with these methods and sometimes even outperforms them.
> >
> > I apologize for the poor wording. I was asking how you explain that the additivity constraint in GNANs does not decrease its performance. Typically, models with the additivity constraint, such as GAMs, NAMs, NodeGAM, and EBM, are outperformed by models without this constraint.

---

> > > ### Comment · Area_Chair_wJum · 2024-08-09
> > >
> > > Thank you very much for checking the response!

---

> > ### Author Response · Authors · 2024-08-13
> >
> > As the rebuttal period comes to an end, we thank the reviewer for the thoughtful and insightful comments and suggestions. We hope our responses have strengthened your confidence in the novelty and merits of this study and that this will be reflected in your final scores.

---

> ### Author Response · Authors · 2024-08-11
>
> We thank the reviewer for the comments and additional questions.
>
> 1. The reviewer suggests adding information on the number of trials conducted for each model. We thank the reviewer for the suggestion.
> This information is provided in lines 272-276 in the main paper.
>
> 2. The reviewer raises a valid point that “traditional GAMs” can be fitted with confidence intervals and thus support hypothesis testing, whereas NAMs do not possess this capability. Since GNAN is an adaptation of NAM, it prioritizes interpretability but does not inherently support hypothesis testing without additional methods, such as bootstrapping. This is an important observation, and we will clarify this in the camera-ready version of the paper. However, we would like to emphasize that our primary claim is that GNAM makes the *model* interpretable, rather than providing an explanation of the *data* itself. As highlighted in [1], these are distinct concepts. For this reason, we do not claim that GNAN reproduces any underlying structure of the data, nor do we compare it to such structures. Instead, GNAN is transparent in its decision-making process, making it valuable for debugging models, identifying biases, and gauging whether the model is trustworthy. We do not claim that it can be used, in its current form, for causal analysis. To prevent any ambiguity, we will explicitly mention this in the camera-ready version.
>
>
>
>
> 3. Regarding the comment that “the paper simply displays some graphics of GNAMs’ predictions,” we would like to clarify that the plots presented in the paper are representations of the model itself, not just “graphics of its predictions.” \
> The predictions made by the model for *any* point can be directly computed from these graphs without requiring any additional information. To the best of our knowledge, no post-hoc method shares this property. Post-hoc models provide explanations of a model’s behavior, but they do not make the model as transparent as GNAN does. In terms of accuracy, post-hoc methods inherit the accuracy of the underlying predictive model they explain, such as GIN. Our experiments demonstrate that GNAN is comparable in accuracy to these models while also offering interpretability. \
> \
> These surprising high accuracies were acknowledged by the reviewer since the additive constraint in GNAM did not lead to a reduction in performance. We hypothesize that this reflects some underlying properties of common graph learning tasks, as was noted by previous works [2,3,4] and discussed lines 283-288.
>
>
> [1] True to the model or true to the data? Chen, H., Janizek, J. D., Lundberg, S., & Lee, S. I., 2020. \
> [2]  Simplifying Graph Convolutional Networks, Wu et al, 2019. \
> [3] A Fair Comparison of Graph Neural Networks for Graph Classification, Errica et al, 2022. \
> [4] Graph Neural Networks are Inherently Good Generalizers: Insights by Bridging GNNs and MLPs, Yang et al, 2023.

---

### Official Review · Reviewer_vkt8 · 2024-07-01

**Soundness:** 3
**Presentation:** 2
**Contribution:** 3
**Rating:** 5
**Confidence:** 3

**Summary:**

This paper introduces the Graph Neural Additive Network (GNAN), a novel interpretable graph neural network based on Generalized Additive Models. GNAN is designed to be fully interpretable through visualizations that clearly demonstrate how it uses relationships between the target variable, features, and graph structure. The paper shows that GNAN performs comparably to traditional black-box GNNs, making it suitable for critical applications where both transparency and accuracy are essential.

**Strengths:**

1. The paper creatively extends Generalized Additive Models to graph neural networks, demonstrating performance comparable to mainstream graph neural networks.
2. The paper conducts extensive experiments to validate the interpretability of GNAN.

**Weaknesses:**

1. The formulas and methods section of the paper is somewhat rough and could be further optimized.
2. GNAN requires calculating the shortest paths and relationships between any two nodes, which could be prohibitively costly for large datasets.
3. Graphs are complex data types, and merely capturing node-level relationships might not replace previous subgraph-level explanation schemes. The paper lacks further analysis and argumentation on this point.

**Questions:**

Check the above weaknesses.

**Limitations:**

NA.

---

> ### Author Rebuttal · Authors · 2024-08-03
>
> We thank the reviewer for the thoughtful comments and encouraging remarks.
>
> ## Weakness 1:
> The reviewer suggests improving the presentation of formulas in the methods section. We thank the reviewer for this suggestion, and we use it to improve the camera-ready version.
>
> ## Weakness 2:
> The reviewer mentions that calculating shortest paths between any two nodes can be costly for large graphs. This is indeed true, but this limitation is not unique to GNAN, and remedies for this problem exist.\
> First, note that pre-computing the shortest paths between nodes is a common practice in the GNN literature as a way of improving the expressive power of the model. For example, in the shortest-path GNN [1], the authors apply message-passing over shortest paths in graphs. In [2], the shortest path length is used as a positional encoding for the graph transformer. These approaches inspired us to utilize shortest-path information to incorporate the graph into the GAMs framework.\
> Second, in cases where the cost of computing all shortest paths is prohibitive, a natural remedy is to clip distances greater than a threshold D. This can be used in many scenarios as it makes sense that remote nodes should have a small influence on each other.\
> Importantly, the distance computation is only required once, and the determined neighborhoods can subsequently be reused at no additional cost. Hence, in some cases, this cost can be amortized over multiple runs on the same graph or executed as a preprocessing step that does not affect the online running time of the model.\
> We demonstrate in the paper several applications to tasks where graph sizes allow the computation of shortest paths. Additionally, we wish to note that GNAN is the first approach to extend GAMs into graph learning, and we believe it opens up many new directions for future research, including finding more efficient methods.
>
>
> ## Weakness 3:
> The reviewer claims that using only node-level information may not be sufficient to replace subgraph-level explanation schemes and that further analysis is required on that matter. We thank the reviewer for this important note. It is true that there may be cases where subgraph-level information is crucial for accurate predictions. Surprisingly, this was not the case in the many datasets we experimented with. \
> Moreover, it is important to note that existing subgraph-level explanations are post-hoc explanations used with black-box GNNs. In contrast, GNAN is an inherently interpretable white-box model rather than a post-hoc explanation. As discussed in the paper in detail in lines 31-44 and 114-125, post-hoc explanations over black-box models are not suitable for high-stake applications where transparency is crucial, such as in healthcare and criminal justice. Therefore, while post-hoc subgraph explanations may be useful in some cases, they are not suitable in many domains. We claim that GNAN may serve this unmet need as it provides full transparency for decision-makers in such applications.
>
>
> [1] Shortest Path Networks for Graph Property Prediction, Abboud wt al. LOG 2022. \
> [2] GRPE: RELATIVE POSITIONAL ENCODING FOR GRAPH TRANSFORMER, Park et al. ICLR 2022.

---

> > ### Comment · Area_Chair_wJum · 2024-08-07
> >
> > Thank you for the response. Dear reviewer vkt8, could you check whether the authors addressed your concerns? Thank you!

---

> > ### Author Response · Authors · 2024-08-13
> >
> > As the rebuttal period comes to an end, we thank the reviewer for the thoughtful and insightful comments and suggestions. We hope our responses have strengthened your confidence in the novelty and merits of this study and that this will be reflected in your final scores.

---

### Official Review · Reviewer_5GdN · 2024-07-04

**Soundness:** 2
**Presentation:** 2
**Contribution:** 1
**Rating:** 4
**Confidence:** 3

**Summary:**

The paper presents the GNAN, a model designed to integrate the interpretability of Generalized Additive Models with GNNs. GNAN aims to address the black-box nature of traditional GNNs by providing explanations through visualization. The model achieves this by learning shape functions for each feature and linearly combining them. Distance functions are used to capture the graph structure influence among nodes. GNAN allows the relationships between the target variable, features, and graph topology to be easily understood. GNAN matches the performance of existing GNNs on several datasets for both node and graph predictions.

**Strengths:**

- Novelty: the idea of combining the GAM and GNN is novel to me.
- Interpretability: As a graph ML model, GNAN provides inherent interpretability, allowing users to understand the model's decision-making process without post hoc explanations
- By avoiding iterative message-passing, GNAN reduces computational bottlenecks and makes parallel computing easier.

**Weaknesses:**

- Novelty of model design: the current model design is not too novel to me. My understanding is that GNAN is a special type of graph transformer, where the positional encoding (dist in the paper) is explicitly combined with the node features using a heuristic function, also attention is removed. However, the high-level idea of decoupling GNN into node feature and positional feature and encoding them separately is the same as graph transformers

- Empirical performance: this is the biggest weakness of the paper to me. For all the datasets, the GNAN model doesn't seem to be better than the baselines

- Presentation: model debugging is claimed as a major contribution, but from the current writing how exactly that can be done is not clear to me. Also, the "inteligibility" in the title and section 4 might be overclaiming. Overall, the model is only for node and graph prediction.

**Questions:**

- Regarding my weakness 1, how is the idea of GNAN different from transformer? I could misunderstood, so maybe the authors can explain more.

- Regarding my weakness 3, How exactly can the method be used for debugging? In line 209 - 215, this point was vaguely discussed as using visualizations to correct misalignment. How can that correction be performed exactly?

---

> ### Author Rebuttal · Authors · 2024-08-03
>
> We thank the reviewer for the thoughtful comments and encouraging remarks.
>
> ## Weakness 1:
> The reviewer claims that GNAN is a special type of transformer and asked for clarifications in Question 1. We think there might be some confusion here that we are happy to clarify. We do not see GNAN as an instance of a transformer. \
> The fundamental idea in GNAN is that each feature is processed separately by the network using a univariate shape function. The values computed by these shape functions are sum-pooled only in the last layer. This is in contrast to transformers, where each layer may compute functions that mix arbitrary numbers of features. GNANs also restrict the way distance and features interact, which allows for their interpretability. \
> While it is true that both GNANs and transformers allow information propagation between any pair of nodes, they are considerably different, as evidenced by the interpretability of GNANs compared to the black-box nature of transformers.
>
>
> ## Weakness 2:
> The reviewer commented that GNAN does not outperform other baselines. We would like to emphasize that the main goal of GNAN is to provide interpretable model for graph learning. The interpretability requirement restricts the type of models that can be used but surprisingly,  we show that in many settings, there is no tradeoff between explainability and accuracy since GNAN is comparable to the commonly used (non-explainable) models, and in some cases, it is even better. As we mentioned in the paper, e.g., line 284, this is non-trivial as it is usually assumed that interpretability comes at the cost of accuracy. Second, we note that GNAN did outperform the tested baselines on the Tolokers, ogb-arxive,  and alpha, mu, and alpha-homo tasks.
>
> ## Weakness 3:
> The reviewer asks how debugging is done. We thank the reviewer for the question. GAMs, including GNAN, allow for debugging and adjustment of the model in several ways. As mentioned in lines 210-215, because GNAN is transparent and can be fully visualized, this allows for several debugging methods. \
> One example given in lines 210-215 suggests using the visualizations of the model for model selection. Specifically, one can select models not only based on accuracy but also based on their alignment with prior knowledge. \
> The visualizations of GAMs also enable debugging biases in the model, which can be addressed in different ways [1, 2, 3]. \
> Additionally, the ability to visualize exactly what the model has learned and observe whether it "makes sense" can help detect code bugs. For instance, we used this property to hunt bugs during the development of the model.
>
>
>
> [1] Interpretability, then what? editing machine learning models to reflect human knowledge and values, Wang et al., 2021. \
> [2] How interpretable and trustworthy are gams, Chang et a.l, 2021. \
> [3] Gam changer: Editing generalized additive models with interactive visualization, Wang et al. 2021.
>
> ## Question 1:
> The reviewer asks how GNAN is different from a transformer. This is addressed in Weakness 2
>
> ## Question 2:
> The reviewer asks how the visualization of GNAN can be used for debugging. This is addressed in Weakness 3.

---

> > ### Comment · Area_Chair_wJum · 2024-08-07
> >
> > Thanks to the authors for their response. Dear reviewer 5GdN: Could you clarify whether the authors addressed your points, especially regarding novelty and contrast to baselines?

---

> > ### Comment · Reviewer_5GdN · 2024-08-10
> >
> > Thank the authors for their response. However, my concerns are not cleared, and I still cannot vote for acceptance of this paper.
> >
> > For my weakness 1 regarding novelty, yes, I agree GNAN and graph transformers are different in some detailed designs, but like I mentioned in my original comment, the "high-level idea of decoupling GNN into node feature and positional feature and encoding them separately is the same as graph transformers". In contrast, in which layer the encoded features are pooled is a marginal difference to me.
> >
> > For my weakness 2 regarding performance, GNAN didn't outperform baselines on half of the datasets. Also, its performance on ogbn-arxiv is the best among the considered baselines, but those baselines are not strong according to the public leaderboard. GNAN's performance on ogbn-arxiv won't stand out if it is compared to other stronger models on the leaderboard.

---

> > > ### Author Response · Authors · 2024-08-11
> > >
> > > Thank you for your insightful comments. We would like to emphasize that the primary objective of this study is to introduce interpretability to learning on graphs. This goal is increasingly important given legislative efforts, such as the EU AI Act, which mandates interpretability in high-stakes applications. Without interpretable models, the use of graph-based learning in such applications may become legally questionable. Therefore, methods like GNAM are crucial to ensuring the relevance of this field in critical areas.
> > >
> > > Regarding the similarity to graph transformers, we would like to clarify that GNAM is an interpretable model, whereas graph transformers typically are not. This distinction is significant, even though there may be some shared internal structures. The outcomes of these models differ markedly. As we propose in lines 296-302, the separation between structure and feature is not the only defining characteristic of GNAM. Instead, it is the separation of features from each other that enhances the interpretability of the model.
> > >
> > > The performance of GNAM should be assessed in the context of the added interpretability. This requirement introduces a constraint, which may lead to some expected performance loss. However, it is noteworthy that this loss, at least on the commonly used datasets in the field, is minimal, if present at all.

---

> > ### Author Response · Authors · 2024-08-13
> >
> > As the rebuttal period comes to an end, we thank the reviewer for the thoughtful and insightful comments and suggestions. We hope our responses have strengthened your confidence in the novelty and merits of this study and that this will be reflected in your final scores.

---

### Official Review · Reviewer_3kSH · 2024-07-11

**Soundness:** 4
**Presentation:** 4
**Contribution:** 4
**Rating:** 7
**Confidence:** 5

**Summary:**

The paper proposes an interpretable by design model for graph data. The proposed model GNAN builds upon Generalised additive models and learns node representations as a distance function and feature shape functions explicitly and independently for each function. Interpretability is then offered by means of visualising the distance and feature shape functions.

**Strengths:**

The paper presents a novel (to the best of my knowledge) yet simple idea of extending generalised additive models to achieve interpretability in learning from graph structured data. It provides fresh perspective on the problem.  It not only departs from the common definitions of explanations for graph data but also from the much hyped GNNs. The paper is very well written and easy to follow. If there are limitations to the current method for large graphs and large feature spaces it present a very valid first step which would be picked up the community for further improvements.

**After discussion period**
I agree with the other reviewers that evaluation can still be improved. To reach the excellence level as my previous score indicated, teh paper would need to demonstrate clearly the claims of the paper (via user studies) for example for model debugging.

**Weaknesses:**

It would be problematic to apply this method directly to datasets with very large feature spaces and large graphs. Can one come up with a strategy to initially prune these large spaces as a preprocessing step?

**Questions:**

1. How sensitive are explanations to model initialisations?
2. Could you provide a more concrete example of a local explanation corresponding to a query node in the scenario of node classification?
3. In the scenario of large feature spaces is it possible to have a quantitative metric to conclude that certain features may not be visually inspected.

**Limitations:**

yes

---

> ### Author Rebuttal · Authors · 2024-08-03
>
> We are gratified by the reviewer’s appreciation of the novelty and interest of our work and thank the reviewer for the valuable feedback.
>
>
> ## Weakness 1:
> The reviewer claims it could be problematic to apply GNAN to large graphs with many features and asks for methods to prune the space as a pre-processing step. We thank the reviewer for the question. \
> One approach is to clip distances greater than a predefined threshold. One can also mask distances using other rules rather than one threshold.
> If needed, there are also many methods for reducing the graph size in advance [1, 2]. \
> We also note that GNAN is implemented in a tensor-multiplication formulation as described in Appendix A, which is therefore optimized when using GPUs. \
> Regarding the cases where the number of features is large, any feature-selection method, e.g., variance-threshold or univariate feature selection, can be applied as a pre-processing step to reduce the number of features. We also addressed the case of limiting the number of feature visualizations in our answer to Question 3.
>
> [1] On the Ability of Graph Neural Networks to Model Interactions Between Vertices, Razin et al.,ICML 2023. \
> [2] DropEdge: Towards Deep Graph Convolutional Networks on Node Classification, Rong et al., ICLR 2020.
>
> ## Question 1:
> The reviewer asks how sensitive are the explanations of GNAN to initializations. We thank the reviewer for this important question as it allows us to clarify important aspects of our proposed algorithm. \
> In the literature about explainability, there is a distinction between explaining the model and explaining the data [3]. The explanations in GNAN are explanations of the model. Regardless of the initialization, the explanation will always describe precisely the model. Therefore, the sensitivity of explanations would simply visualize the model’s sensitivity to initializations. \
> Nonetheless, following the reviewer’s question, we visualized the heatmap of the mutagenicity experiment using a GNAN trained with 3 different seeds. The heatmaps are attached in the PDF file allowed in the global comment. The values in the heatmap slightly differ, but the trend is stable.
>
> [3] Chen, H., Janizek, J. D., Lundberg, S., & Lee, S. I. (2020). True to the model or true to the data?. arXiv preprint arXiv:2006.16234.
>
>
> ## Question 2:
> The reviewer asks for a concrete example of a local explanation corresponding to a query node in node classification. We thank the reviewer for the opportunity to address this topic. Given a query graph and node, we highlight other nodes in the graph in a way that corresponds to their contribution to the query node’s prediction based on the learned distance and feature functions.  For example, if we observe from the visualization that neighbors of distance x with feature k contribute towards positive labels, we can highlight such nodes in the given query graph, and the distances are computed with respect to the query node. We will make sure to add such examples to the camera-ready version.
>
> ## Question 3:
> The reviewer asks how to conclude which features to inspect in the case of large feature spaces. We thank the reviewer for this important question. \
> This is a fundamental problem in explainability that is common to many methods. Even a linear model or a tree may lack human interpretability in the context of a large feature space. A plausible method to mitigate the problem is to use regularization terms to induce sparsity in the use of features or use feature selection. \ Another approach is to use a post-processing step of removing the contribution of features for which the shape-function has low variance. \
>  Additionally, in some cases, users have pre-defined features of interest to examine. For example, doctors may be interested in observing the effect of specific genes. \
> We will add this important discussion to the camera-ready version.

---

> > ### Comment · Area_Chair_wJum · 2024-08-07
> >
> > Thanks to the authors for this response. Dear reviewer 3kSH: Could you read the other reviews and check how you regard the paper in light of the other reviews as well as the author response? Thank you!

---

> > ### Author Response · Authors · 2024-08-13
> >
> > As the rebuttal period comes to an end, we are very thankful for the thoughtful and insightful comments, as well as the encouraging words about our study. We hope that the discussion has further strengthened your opinion of our work.

---

> > ### Comment · Reviewer_3kSH · 2024-08-14
> > **Thanks for the response**
> >
> > Thanks for the rebuttal. I maintain my positive score.

---

### Author Rebuttal · Authors · 2024-08-03

We thank the reviewers for their important questions and thoughtful feedback.
Attached is a PDF file for reviewer 3kSH.

---

### Decision · Program_Chairs · 2024-09-25

**Decision:**

Accept (poster)

**Comment:**

The paper proposes a novel architecture for graph neural nets, where the influence of nodes y influencing the representation of node x is regulated by normalizing by the number of other nodes y' at that distance and a learnable distance function. Further, predictions are performed via a simple addition operator, which supports interpretability.

Reviews generally agree that the core architecture is at least somewhat novel and enhances interpretability. It is also apparent from the experiments that it can perform roughly on par with existing methods while simplifying the architecture. The topic of the paper - intrinsically interpretable graph neural nets - is also still relatively fresh with most prior work in GNNs focusing on post-hoc explanations.

There are still some concerns, though:
- Using shortest paths for the computation of representations has been present in the literature before. Similarly, additive models are pre-existing. As such, the degree of novelty in the proposed architecture is, hence, somewhat questionable.
- The interpretability is only evaluated on two example cases and no user study has been provided. As such, the interpretability claims are not fully substantiated by evidence.

Overall, the contribution provided by a strongly simplified model seems interesting enough to a broad part of the NeurIPS community that a presentation as poster is warranted.